# What is needed in a Knowledge Graph Management Platform? A survey and a proposal

Samira Babalou[*] [ID] [1 2], Franziska Zander [ID] [1 2], Erik Kleinsteuber [ID] [1], Badr El Haouni [ID] [1], David Schellenberger Costa [ID] [2], Jens Kattge [ID] [2 4], Birgitta König-Ries [ID] [1 2 3]

[1]Heinz-Nixdorf Chair for Distributed Information Systems
Institute for Computer Science, Friedrich Schiller University Jena, Germany
[2]German Center for Integrative Biodiversity Research (iDiv), Halle-Jena-Leipzig, Germany
[3]Michael-Stifel-Center for Data-Driven and Simulation Science, Jena, Germany
[4]Institute of Biology/Geobotany and Botanical Garden, Martin Luther University, Halle, Germany
corresponding author: samira.babalou@uni.jena.de

**Abstract.** Knowledge Graphs (KGs) play a significant and growing role for semantics-based support of a wide variety of applications. Until recently, creating and maintaining such knowledge graphs was done in a one-off manner requiring significant manual effort and expertise. Over the last few years, the first KG management platforms supporting the lifecycle of KGs from their creation to their maintenance and use have appeared. In this paper, we first survey these platforms. We then take a step further and identify common functionalities across such platforms. We discuss nineteen such functionalities categorized into four groups: creating, extending, using, and maintaining KGs. Based on the findings of this analysis, we present our proposed KG management platform for the biodiversity domain, iKNOW. We focus on the architecture and the KG creation workflow, but also touch on other aspects.

**Keywords:** Semantic Web . Knowledge Graph . Knowledge Graph Platform . Data Services and Functionality

## 1 Introduction

Increasingly, Knowledge Graphs (KGs) form the semantic data management backbone for a wide variety of applications. A KG [1] consists of nodes connected by edges. It is built from on a set of data sources via different techniques. Besides the instances, KGs can also contain schema information, which can be refined or augmented, e.g., by using a reasoner. Assigning unique identifiers to KG's entities can accelerate the interlinking with other resources on the web. The underlying structure of KGs opens a door for further functionalities such as visualization, supporting keyword search and complex queries via a SPARQL endpoint.

Although KGs have widely gained attention in industry and academia, developing and managing their lifecycle requires a huge effort, expertise, and different functionalities. While, in the beginning, KGs were typically one-off manual efforts, there is a growing awareness that to exploit the capabilities of Knowledge Graph technologies to the maximal extent, support for their creation, access, update, and maintenance is needed. Many of these functionalities are not specific to any given KG but can be provided rather generically. KG platforms aim to do just that.

As our contribution, in this paper, we survey existing KG management platforms and compare them in a general way. We then take a step further and analyze nineteen functionalities in four categories: creating, extending, using, and maintaining KGs. To the best of our knowledge, this is the first survey about KG platforms. Based on the findings of this survey and the needs in our domain, biodiversity research, we have designed our own KG platform. We present this platform, iKNOW, in the second part of the paper.

The rest of the paper is organized as follows. Section 2 surveys existing KG management platforms. The common functionalities of platforms are discussed in Section 3. Our proposal for a KG management platform focussed on biodiversity, iKNOW, is presented in Section 4. The paper is concluded in Section 5.

## 2 Literature Review

In this paper, we define a *Knowledge Graph Platform* as a web-based platform for creating, managing, and making use of KGs. Such platforms mostly cover the whole lifecycle of KG application and include relevant services or functionalities for interaction and management of KGs.

We contrast these from efforts to build an individual, specific KG. There have been many such efforts in different domains: e.g., Ozymandias [2] in the biodiversity domain, BCKG [3] in the biomedical domain, and I40KG [4] in the industrial domain. These KGs were built one time, and now their associated websites provide the KG access and usage. Such approaches are out of the scope of this paper. Rather, we focus on KG management platforms, which offer a set of operations such as generation and updates on the KG.

In the following subsections, we first present the survey methodology used in this paper, then we briefly summarize the existing KG management platforms and compare them in a general way.

### 2.1 Survey Methodology

In this subsection, we describe our systematic approach to finding publications on KG platforms: We have queried for the keyword "Knowledge Graph Platform" in the Google Scholar search engine [1]. At the time of querying, this resulted in 162 papers (including citation and patents). We used Publish or Perish 8 tool [2]

---

[1] https://scholar.google.com/ access on 09.02.2022
[2] https://harzing.com/blog/2021/10/publish-or-perish-version-8

to save the result of the query. The result is available in our GitHub repository [3]. Among the list of papers, we selected the relevant papers manually. We aimed to select papers that focus on the KG management platform. Some papers appeared in the result of google scholar because our keyword exists in their texts (e.g., in the literature review section), but those papers mainly do not propose a new KG platform. We did not include such cases. Moreover, we did not consider survey papers and papers written in a language other than English. In our repository, we specified which papers have been selected, and for non-selected ones, we clarified the reason. As a result, we came up with 11 KG platforms, briefly detailed in the following sub-section.

## 2.2  Existing KG Management Platforms

In this section, we give a brief overview of existing platforms:

- **BBN (Blue Brain Nexus)** [5] is an open-source platform. The KG in this platform can be built from datasets generated from heterogenous sources and formats. BBN has three main components: i) Nexus Delta, a set of services targeting developers for managing data and knowledge graph lifecycle; ii) Nexus Fusion, a web-based user interface enabling users to store, view, query, access, and share (meta)data and manage knowledge graphs; and iii) Nexus Forge, a Python user interface enabling data and knowledge engineers to build knowledge graphs from various data sources and formats using data mappings, transformations, and validations.
- **CPS** (Corpus Processing Service) [6] is a cloud platform to create and serve Knowledge Graphs over a set of corpus. It uses state-of-the-art natural language understanding models to extract entities and relationships from documents.
- **HAPE** (Heaven Ape) [7] is a programmable KG platform. The architecture of HAPE is designed in three parts: the client-side, which provides various kinds of services to the users; the server-side, which provides different knowledge management and processing, and the third part, which is KG's knowledge base. The applicability of the platform has been shown over DBpedia data. Moreover, the quality of created KG has been evaluated via metrics introduced in [8]. Although the authors in their published paper claimed that the platform is open to the public, to the best of our knowledge, there is no link to the platform source code or the online web portal.
- **Metaphactory** [9] is an enterprise platform for building Knowledge Graph management applications. This platform supports different categories of users (end-users, expert users, and application developers), has a customizable UI, and enables the rapid building of use case-specific applications. Metaphactory allows configuring and managing connections to many data repositories. In this platform, data sources are virtually integrated with an ontology-based data access engine, i.e., on-the-fly integration of

---

[3] https://github.com/fusion-jena/iKNOW

diverse data sources. The platform is assessed via assessment parameters introduced in [10].

- **Meng et al**., [11] proposed a power marketing KG platform. The authors used a Machine Learning (ML) method to extract knowledge from unstructured text. The knowledge instances are stored in relational data. The relationship of knowledge is stored in a graph database.
- **MONOLITH** [12] is a KG platform combined with Ontology-based Data Management (OBDM) capabilities over relational and non-relational databases to result in one (virtual) data source. The functionalities provided by MONOLITH can be split into two groups: one dedicated to managing OWL ontologies and providing OBDM services, exploiting the mappings between ontology and database; the other to managing KGs and providing services over them. These two groups are linked together, allowing to build the KGs through semantic data access from the results of the ontology queries.
- **News Hunter** [13] is geared towards supporting journalism by aggregating and semantically integrating news from a variety of sources. It is based on a microservices architecture and consists of a number of independent such services: First, an extensible set of harvesters are aggregated from information from individual sources or existing news. Harvested news items and relevant metadata are deduplicated and stored in a source database. A translator converts items into a canonical language; this allows for cross-language news linking and the application of the broad range of existing NLP (Natural Language Processing) tools. This step, called Lifting in the paper, runs the extracted news items through an NLP pipeline which performs named-entity recognition as well as sentiment and topic analysis. Results of this step are stored in a graph database. ML-based classifiers are used to assign labels to news items thereby annotating them with terms from a common ontology modeling. Via an enricher, the KG can be augmented by information from external sources, e.g., DBpedia Spotlight.
- **TCMKG** [14] is a KG platform for Traditional Chinese Medicine (TCM) based on the deep learning method. First, an ontology layer represents the knowledge-based diagnosis and treatment process. It includes core entities of the domain with their associated relations. Then, with the help of a named entity recognition (NER) model, TCM entities from unstructured data are extracted.
- **UWKGM** [15] is a modular web-based platform for KG management. It enables users to integrate different functionalities as RESTful API services into the platform to help different user roles customize the platform as needed. The platform consists of three main components: the backend (API), the frontend (UI), and the system manager (for installation, upgrading, and deployment). The embedded entity suggestion module enables automatic triple extraction and maintains human involvement for quality control.
- **YABKO** [16] is the successor of HAPE and aims to support the life cycle research on KGs. Researchers can upload their KGs and tools to the YABKO platform that can be free of use for other researchers' experiments. For

any requested experiment, YABKO assigns necessary resources (space, time, KGs, tools) to it. After finishing an experiment, the short-term experiment will be dissolved, while the long-term ones can continue to exist on the condition of publishing their results. The core motivation of building YABKO is to help visitors use open-source techniques and resources to perform experiments on KGs and share experiences with other researchers.

- **Yang et al.**, [17] proposed a cloud computing cultural knowledge platform over multiple data sources such as Chinese Wikis, lexical databases, and cultural websites. The platform restricts the knowledge in the field of Chinese public cultural services instead of common sense knowledge. The platform has a set of services for building, updating, and maintaining the KG. It uses rule-based reasoning methods to analyze the existing KG relations to predict the new possible relations.

### 2.3 Comparing Existing KG Management Platforms

In Table 1, we summarize general information about the introduced KG platforms with respect to: their *Name*, the *Year* of release (based on the published paper), the used *Source Data Type* to build KGs, their target applications in industry or *Academia*, their *Open-Source* accessibilities, the availability of an *Online Demo*, a test with a *Use Case Study*, and, finally, the supported *KG Construction Method* by the platform. Looking at the table, one can observe that:

- most platforms have been introduced in the last three years. This shows that the field is still young and most likely still evolving. This observation is confirmed by our analysis of provided functionality (see below).
- the platforms are very heterogeneous with respect to the number and type of data sources they support.
- for KG construction, basically, all platforms follow an ETL (Extract, Transform, Load) process along with Machine Learning (ML) approaches. They differ in how adaptable this process is and, partially depending on the type of supported data sources, on the concrete steps involved in this process.
- a (to us) surprisingly high percentage of platforms are designed for use within industry (as opposed to academia). This may be one of the reasons why quite many of these platforms are not open source.
- all platforms had a use case study to show the capabilities of the platform by describing a specific KG's usage in a selected application domain.

## 3 Common Functionalities in KG Management Platforms

In this section, we take a closer look at the KG platforms, extract what functionalities they offer and compare them with respect to these functionalities. We consider a functionality for a platform if the functionality is mentioned in the

Table 1: Comparing existing KG management platforms concerning their names, the year of release, the type of source data used to build KGs, targeting academia or not, being open-source, availability of an online demo, testing in a use case study, and the KG construction method. $\checkmark^*$ means currently not available and - shows not mentioned.

| no. | Platform | Year | Source Data Type | Academia | Open-Source | Online Demo | Use Case Study | KG Construction Method |
|---|---|---|---|---|---|---|---|---|
| 1 | BBN [5] | 2021 | different types | $\checkmark$ | $\checkmark$ | $\checkmark$ | $\checkmark$ | customized ETL process |
| 2 | CPS [6] | 2020 | text | $\times$ | $\times$ | $\times$ | $\checkmark$ | Machine Learning |
| 3 | HAPE [7] | 2020 | different types | $\checkmark$ | $\times$ | $\times$ | $\checkmark$ | - |
| 4 | Metaphactory [9] | 2019 | different types | $\times$ | $\times$ | $\checkmark$ | $\checkmark$ | customized ETL process |
| 5 | Meng et al [11] | 2021 | unstructured text | $\times$ | $\times$ | $\times$ | $\checkmark$ | Machine Learning |
| 6 | MONOLITH [12] | 2019 | - | $\times$ | $\times$ | $\times$ | $\checkmark$ | customized ETL process |
| 7 | News Hunter [13] | 2020 | text | - | $\times$ | $\times$ | $\checkmark$ | Machine Learning |
| 8 | TCMKG [14] | 2020 | different types | - | $\times$ | $\times$ | $\checkmark$ | Machine Learning |
| 9 | UWKGM [15] | 2020 | unstructured text | - | $\checkmark$ | $\checkmark^*$ | $\checkmark$ | customized ETL process |
| 10 | YABKO [16] | 2021 | different types | $\checkmark$ | $\times$ | $\times$ | $\checkmark$ | - |
| 11 | Yang et al [17] | 2017 | different types | - | $\times$ | $\times$ | $\checkmark$ | Machine Learning |

respective papers. Platforms may possess other functionalities not mentioned in the papers. So a missing entry does not necessarily mean a platform does not offer certain functionality. Overall, many of the papers were surprisingly vague about what functionality the platforms offer, so that not always a clear decision was possible. From our analysis, we identified nineteen different functionalities which can be grouped into four categories as follows:

- **Functionalities for creating a KG**: The platform can support different functionalities to build the KG with the desired quality:
  - **Data preprocessing** [5,7,14,17]: Before information from a data source can be used in a KG, several preprocessing steps may be needed. These include data cleaning and data transformation in a format suitable for ingestion.
  - **Entity and relation extraction** [6,7,9,13–15,17]: In particular, when creating KGs out of unstructured information like documents, entity and relation extraction can require complex processing. But even for structured data, this step is often necessary.
  - **Schema generation** [7,9,12–14,17]: If a KG is supposed to contain not just a set of instances, but also type information about them, a schema needs to be created.
  - **KG validation** [5, 7, 9, 12, 16, 17]: When a KG combines data from different sources, the initial data cleaning step, which happens at the level of an individual source, may not be sufficient to ensure that the integrated KG is consistent. Thus, the platform may take a further step on quality checking and validation of the KG.
- **Functionalities for extending and augmenting KGs**: This group of functionalities allows for extending KGs with additional information from other sources or from within the KG itself. While cross-linking extends a KG with information provided somewhere else, a variety of techniques are

used to extend KGs "from within". They include reasoning to infer hidden knowledge, KG refinement and the computation of KG embeddings as a basis for link prediction and similarity determination.

- **Cross-linking** [5, 9, 13, 17]: This functionality enables the cross-linking of KG' entities to other resources or KGs like Wikidata or DBpedia. According to the linked open data (LOD) principles [18], each knowledge resource on the web receives a stable, unique and resolvable identifier.
- **KG embedding** [7, 9, 14–17]: This is a popular method in particular for link prediction and similarity detection and can help to uncover hidden information in a KG.
- **KG refinement** [5, 15–17]: In some cases, after checking the quality of the generated KG, a refinement process (e.g., validating the KG to identify errors and correcting the inconsistent statements) can take place.
- **Reasoning** [7, 12, 13, 16, 17]: The reasoning functionality can help more knowledge be inferred in a KG mainly with the help of a reasoner. We consider this as KG augmentation, too.

– **Functionalities for using KGs**: Depending mostly on the targeted user group, platforms can support one or several ways to interact with the created KG:

- **GUI (Graphical User Interface)** [5–7, 9, 11–17]: A GUI in a platform is functionality that eases user interaction with the platform.
- **Visualization** [5, 7, 9, 11, 14, 15, 17]: The platform can provide different types of visualization of the KG to help for better understanding. CPS [6] has a visualization type for building queries, only.
- **Keyword search** [5, 7, 9, 11, 12, 15–17]: This functionality enables searching for a keyword over the developed KG in the platform.
- **Query endpoint** [5–7, 9, 11–14, 16, 17]: In the KG management platform, by a query endpoint functionality, the information over the KG can be queried mostly via SPARQL or using graph queries.
- **Query catalog** [9, 12]: The functionality of having a query catalog in the KG management platform enables to use pre-determined (customized) queries or store the queries for future reuse.

– **Functionalities for maintaining and updating KGs**: Once a KG has been built, it may be desirable to manage access, keep track of provenance, update the KG with new or additional sources, and curate it.

- **Provenance tracking** [5, 6, 9, 13]: The platform can track the provenance of KG's entities. Such functionalities can ease the maintenance and updating the KGs.
- **Update KG** [5, 9, 12, 14, 15]: A KG management platform can have the functionality to update and edit the previously generated KG. After this process, KG validation might be required.
- **KG curation** [5, 9, 15, 17]: The platform can have KG curation functionality that mostly relies on human curation.
- **Different user roles** [5, 7, 9, 11, 12, 15–17]: The platform can have functionality that considers different user roles, such as end-users or expert users. This functionality can support different user groups with different access to the other platforms' functionalities.

- **User management and security** [5–7, 9, 11, 12, 15–17]: This functionality can manage user access based on their roles and check the access level and security over the KG in the platform.
- **Workflow management** [5]: The platform can allow to store and replay the creation workflow that can be re-executed.

Table 2 shows the distribution of the functionalities across the KG management platforms. The functionalities are ordered from top to down based on their frequency of availability on the existing platforms. In the last row, we show the total number of supported functionalities of each platform. From this table, our lessons learned are:

- the functionalities in the "KG creation" category are a necessity; thus, they are covered by most platforms. However, one needs to keep in mind, that the platforms differ significantly in what exactly they offer here. Partly, this depends on the supported source data types (e.g., platforms geared towards building KGs from text typically provide NLP-based entity extraction).
- there is a low effort on developing functionalities regarding the KG maintenance category.
- the graphical user interface is the most supported functionality by all platforms.
- the workflow management is the least supported functionality by the existing platforms.

Overall, the figure quite clearly shows that this is a still young and immature field, where so far, no clear set of commonly offered functionality has evolved. We believe that this will happen over time. Meanwhile, potential users of a platform need to carefully check what their requirements are and whether a given platform meets them.

## 4 Our Proposal: a KG Management Platform in the Biodiversity Domain

Our work is motivated by a strong need for KGs in the Biodiversity Domain identified, e.g., by Page [2] and OpenBiodiv [19]. So far, in biodiversity as in many other domains, the few existing KGs have been created largely manually in one-off efforts. If the potential for KGs is to be leveraged for this important domain, it is our conviction, that a KG management platform providing both generic and discipline-specific (e.g., dealing with species) functionality is needed that allows Low-Code (or even No-Code) development, maintenance, and usage of KGs. Using such technologies will reduce the barriers for non-semantic web experts to use and finally benefit from KGs to explore new exciting findings.

The iKNOW project [20] aims to create such a platform, built around a semantic-based toolbox. The project is a joined effort by computer scientists and domain experts from the German Centre for Integrative Biodiversity Research

Table 2: Distribution of functionalities with respect to existing KG management platforms. The functionalities are ordered from top to down based on their frequency of availability on the existing platforms. The last row shows the number of supported functionalities of each platform.

| | BBN | CPS | HAPE | Metaphactory | Meng et al | MONOLITH | News Hunter | TCMKG | UWKGM | YABKO | Yang et al |
|---|---|---|---|---|---|---|---|---|---|---|---|
| Graphical User Interface | ● | ● | ● | ● | ● | ● | ● | ● | ● | ● | ● |
| Query endpoint | ● | ● | ● | ● | ● | ● | ● | ● | ○ | ● | ● |
| User Management & security | ● | ● | ● | ● | ● | ● | ○ | ○ | ● | ● | ● |
| Different user roles | ● | ○ | ● | ● | ● | ● | ○ | ○ | ● | ● | ● |
| Keyword search | ● | ○ | ● | ● | ● | ● | ○ | ○ | ● | ● | ● |
| Visualization | ● | ○ | ● | ● | ● | ○ | ○ | ● | ● | ○ | ● |
| Entity & relation extraction | ○ | ● | ● | ● | ○ | ● | ● | ○ | ○ | ○ | ● |
| KG validation | ● | ○ | ● | ● | ○ | ● | ○ | ○ | ○ | ● | ● |
| Schema generation | ○ | ○ | ● | ● | ○ | ● | ● | ● | ○ | ○ | ● |
| Embedding | ○ | ○ | ● | ● | ○ | ○ | ○ | ● | ● | ● | ● |
| Reasoning | ○ | ○ | ● | ○ | ○ | ● | ● | ○ | ○ | ● | ● |
| Update KG | ● | ○ | ○ | ● | ○ | ● | ○ | ● | ● | ○ | ○ |
| KG curation | ● | ○ | ○ | ● | ○ | ○ | ○ | ○ | ● | ○ | ● |
| KG refinement | ● | ○ | ○ | ○ | ○ | ○ | ○ | ● | ● | ● | ● |
| Provenance tracking | ● | ● | ○ | ● | ○ | ○ | ● | ○ | ○ | ○ | ○ |
| Cross-linking | ● | ○ | ○ | ● | ○ | ○ | ● | ○ | ○ | ○ | ● |
| Data preprocessing | ● | ○ | ● | ○ | ○ | ○ | ○ | ● | ○ | ○ | ● |
| Query cataloge | ○ | ○ | ○ | ● | ○ | ● | ○ | ○ | ○ | ○ | ○ |
| Workflow mangement | ● | ○ | ○ | ○ | ○ | ○ | ○ | ○ | ○ | ○ | ○ |
| Number of supported functionalities | 14 | 5 | 12 | 15 | 6 | 11 | 7 | 8 | 9 | 9 | 15 |

(iDiv) [4]. The work benefits from the wealth of well-curated data sources and expert knowledge on their creation, cleaning, and harmonization available at iDiv. Thus, for now, iKNOW focuses on the (semi-)automatic, reproducible transformation of tabular biodiversity data into RDF statements. It also includes provenance tracking to ensure reproducibility and update ability. Further, options for visualization, search, and query are planned. Once established, this platform will be open-source and available to the biodiversity community. Thus, it can significantly contribute to making biodiversity data widely available, easily discoverable, and integrable.

## 4.1 Workflow in the KG Creation Scenario

After the quite abstract high-level description of iKNOW above, let us now take a closer look at one key functionality, the creation of a new KG. In this paper, we view Knowledge Graph generation as a construction process from scratch,

---

[4] https://www.idiv.de/en/index.html

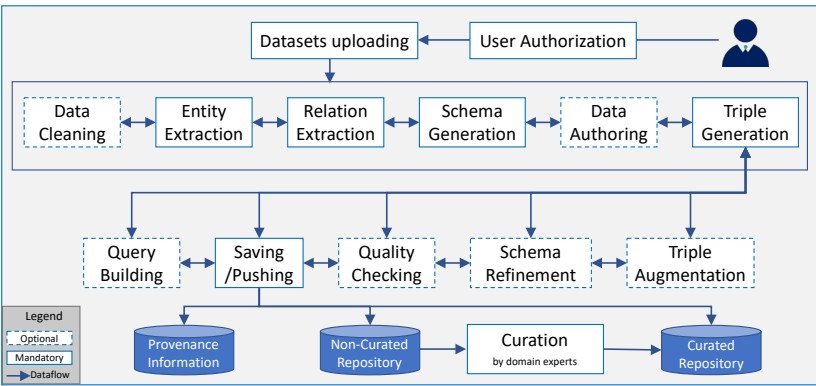

Fig. 1: Workflow in the KG Creation Scenario at iKNOW.

i.e., using a set of operations on one or more data sources to create a Knowledge Graph.

Figure 1 shows the planned iKNOW workflow for the KG creation scenario. It is a generalized one based on the existing platforms. The workflow shows the data flow between the steps towards KG generation. Not all steps are mandatory; some optional processes in each step can add further value to the KG based on the user's needs.

For every uploaded dataset, we build a sub-KG. It will be the subgraph of the main KG in iKNOW. In the first step, users go through the authentication process. The verified users can upload their datasets. If required, the data cleaning process will take place. We offer different tools for this step, which users can select and adjust based on their needs. As we observed, most uploaded data in iKNOW are well-curated, so not all datasets might require this step. For this reason, we consider it as an optional step.

In the *Entity Extraction* step, we map the entities of the dataset to the corresponding concepts in the real world (which build instances of sub-KGs). This mapping is the basis for interlinking entities with external KGs like Wikidata or domain-specific ones. Each mapped entity is a node in the KG. For this process, we have embedded different tools at iKNOW, in which users can select the desired tool along with the desired external KGs.

In the *Relation Extraction* step, the relations between the KG's nodes will be extracted via the user-selected tool. Note that in the entity and relation extraction steps, the tools return the extracted entities and relations to the user. Through our GUI, the user can edit them (*Data Authoring* step).

Each column from the relational dataset refers to a category in the world. We consider the types of the column as classes in the KG. Along with the extracted

relations in the previous step, the schema of this sub-KG will be created in the *Schema Generation* step.

In the *Triple Generation* step, (subject, predicate, object)-triples based on the extracted information from the previous steps will be created. Note that, nodes in the KG are subjects and objects, and relationships are predicates. The triples are generated for classes and instances in the sub-KG.

After these processes, the generated sub-KG can be used directly. However, one can take further steps such as: *Triple Augmentation* (generate new triples and extra relations to ease KG completion), *Schema Refinement* (refine the schema, e.g., via logical reasoning for the KG completion and correctness), *Quality Checking* (check the quality of the generated sub-KG), and *Query Building* (create customized SPARQL queries for the generated sub-KG).

In the *Pushing* step of our platform, the generated KGs are saved first at a temporal repository (shown by "non-curated repository" in Figure 1). After a manual data curation by domain experts in the *Curation* step, the KG will be published in the main repository of our platform. With this step, we aim to increase the trust and correctness of the information on the KG.

All information regarding the user-selected tools with parameters and settings along with the initial dataset and intermediate results will be saved in every step of our platform. With the help of this, users can redo the previous steps (which shows by arrows in both directions). Moreover, this enables us to track the provenance of created sub-KG. In each step mentioned above, we plan to have a tool-recommendation service to help the user select the right tool for every process. For that, we will consider different parameters, such as the characteristics of the dataset and tools.

### 4.2 iKNOW Architecture

Figure 2 shows the planned architecture of iKNOW in five layers:

- In the **User Administration** layer, access level and security will be controlled. Authorized users can generate or update the KG. All end-users can search and visualize the KG. The platform's admin can add new tools or functionalities and approve the user registration. The KG curator curates the recent changes on the KG (newly added sub-KG or updates on previous information on KG).
- The **Web-based UI** layer shows different scenarios for KG management: building a KG, updating the KG, visualizing the KG's triples, and keyword and SPARQL search.
- The **Platform Services** provides a set of required services for the KG management functionalities.
- The **Data Access Infrastructure** manages the communication of services and data storage.
- At the bottom level of the iKNOW platform, the **Data Storage** layer contains the graph database repository (triple management), provenance information, and user information management.

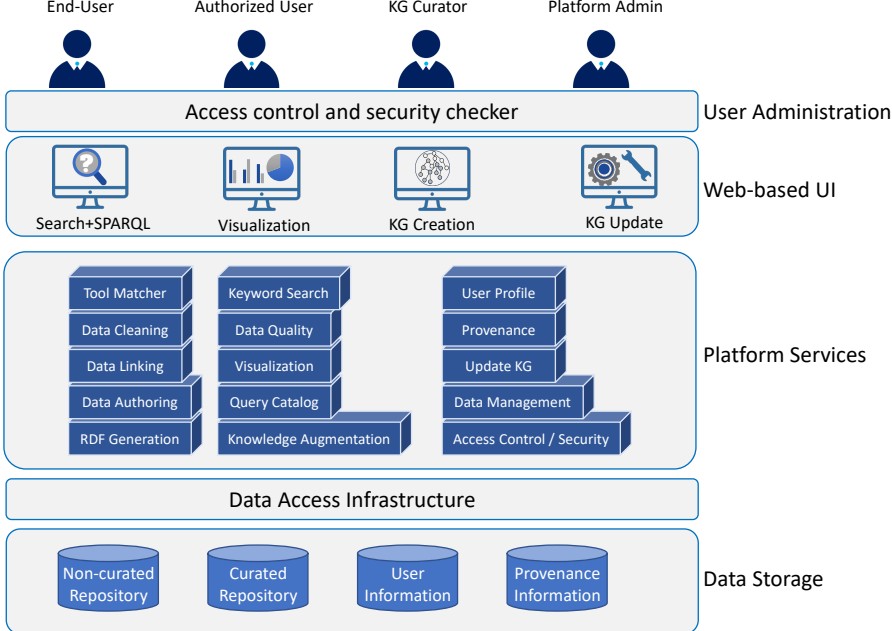

Fig. 2: Architecture of iKNOW in five layers.

### 4.3 Implementation

The iKNOW platform is currently under development (`https://planthub.idiv.de/iknow`). The Python web framework Django [5] is used for the backend with a PostgreSQL [6] database to maintain users, services, tools, datasets, and the KG generation parameters in the iKNOW platform (used in provenance tracking). We use the compiler Svelte [7] with SvelteKit as a framework for building web applications to create a user-friendly web interface. For security, maintenance, and provenance reasons, all tools from external providers used within the workflow will be executed in a sandbox using Docker [8]. For managing the triplestore, we are using the graph database Blazegraph[9]. Any sub-KG created by an end-user, first, will be placed at the non-curated triplestore. After curation by domain experts, the new sub-KG will be added to the curated triplestore. The curated triplestore also serves as the base for SPARQL queries and the keyword search via search engine Elasticsearch [10].

---

[5] `https://www.djangoproject.com`

[6] `https://www.postgresql.org/`

[7] `https://svelte.dev/`

[8] `https://www.docker.com/`

[9] `https://blazegraph.com/`

[10] `https://www.elastic.co/elasticsearch/`

iKNOW is a modular platform, which increases the flexibility of our platform and allows adding new tools. Our ultimate goal is to provide a large set of tool choices for the end-user. Although only a few tools are embedded so far, we plan to add more tools for each functionality in the platform. Then users have a variety of choices with respect to different needs and use cases. Our open-source code and modular designs of our platform make both the front and backend of our platform easily extendable. We encourage users (new developers) to use or extend our reusable UI components to speed up their development.

## 5    Outlook

In this paper, we surveyed eleven KG management platforms and provided a general view of their differences on the used data sources, KG construction approaches, and availability. Taking a closer look, we identified nineteen functionalities offered by one, several or all of these platforms and categorized them into four groups along the lifecycle of a KG. We observed that none of the surveyed platforms supports all of the functionalities. The only category that all platforms strongly support is creation of KGs. Beyond that, so far, there seems to be no agreement on a core set of functionalities. Even within the "creation" category, approaches vary a lot. Partly, this can be attributed to the data source types or user groups targeted by a platform. This, together with the fact that many of the platforms are not open source and/or not available so far limits the choice of platform potential users have. They need to check very carefully whether a specific platform matches their needs.

We did this analysis for our domain, biodiversity research. As a result, we presented our proposed platform, iKNOW.

We conclude that further, domain-specific platforms (or domain-specific extensions of general platforms) are needed to fully leverage the power of KGs across domains. We also recommend, that platform developers should strive to support KGs along their lifecycle beyond just the creation stage. We do believe that both developments will occur as the field matures.

## Acknowledgements

The work described in this paper is conducted in the iKNOW Flexpool project of iDiv, the German Centre for Integrative Biodiversity Research, funded by DFG (Project number 202548816). It is supported by iBID, iDiv's Biodiversity Data and Code Support unit. We thank our college Sven Thiel for comments on the manuscript.

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
