# OpenReview forum: "What is needed in a Knowledge Graph Management Platform? A survey and a proposal"
_kg-construct.github.io/KGCW/2022/Workshop — Submitted to KGCW 2022_

### Official Review · ~Umutcan_Simsek1 · 2022-03-31
**a survey of knowledge graph management platforms, without any explanation of the impact of the biodiversity domain and an immature proposal**

**Rating:** 5
**Confidence:** 5

**Review:**

The authors present a survey of knowledge graph management platforms and make a proposal of their own platform. The paper analyses quite number of systems and describes their own (rather early stage) implementation of a knowledge graph management system. The systems analyzed are representative (except YABKO seems to be a meta-platform) for both academic and industrial landscape. However, there are two major points that pushes the paper below the acceptance threshold:
1. The paper looks like two short papers crammed into one long paper: The survey alone with the requirements stemming from the biodiversity domain would have been a very good paper with a proper discussion and future work. The proposed system could have been described more briefly, based on the lessons learned since even in this longer version the descriptions of almost all modules are quite superficial and even the well explained parts raise some technical questions. For example, the generated subgraphs are processed in isolation, but it is not clear what happens if they interact with each other (e.g. talk about the same thing and have different values for the same property). Also it is not clear what is the impact of biodiversity domain on the implementation of this system.
2. The requirements in the survey framework are defined rather chaotically and not completely fit to the motivation of the paper: The paper explicitly specifies (e.g. second-to-last paragraph in the Outlook section) that the analysis is done for the biodiversity domain. However, the requirements are obtained from the analysis of the 11 systems and are rather generic. Which is not a bad thing on its own, but contradicts with the motivation of the paper. The requirements are presented chaotically in the sense that some notions are under-specified and have no clear distinction between them. For instance, it is not clear what steps the knowledge graph lifecycle contains and how they relate to the 4 requirement categories. Moreover, it is not clear what the difference between KG validation and refinement is and what KG curation really contains.

Due to my points above, I believe the paper is below the acceptance threshold, however it has potential to be a good survey and a good tool paper in the future.

---

### Official Review · ~Mario_Scrocca1 · 2022-04-01
**A survey and a proposal for Knowledge Graph Management Platforms to be further refined**

**Rating:** 5
**Confidence:** 4

**Review:**

The paper is divided into two parts describing (i) a survey of Knowledge Graph Management Platforms from the literature (ii) the iKNOW platform developed by the authors for the biodiversity domain.

The survey explores a wide set of Knowledge Graph Management Platforms describing them briefly and identifying the set of features implemented by each platform. I appreciate the insights highlighted and the categorized list of extracted functionalities. However, I think that the description of the surveyed platforms is limited and heterogeneous and could have been extended and made more systematic to improve the contribution. For example, Table 1 contains very limited information for each column and it could have been relevant to describe for each platform: the “different types” of source data considered, the domain/some details of the use case considered, the distinctive characteristics of the ML/customised ETL process implemented, etc.

The second part of the paper describes the iKNOW platforms proposed by the authors but it is not well integrated with the first one. The authors describe a complete “KG Creation Scenario” but it covers a broader set of functionalities with respect to the "create" group of functionalities defined previously (also from extend, use and maintain groups). In general, the paper seems divided in two distinct parts because the presented iKNOW platform is not “framed” adopting the methodology of the survey and not compared with the other platforms. Furthermore, the authors claim that the iKNOW platform is designed considering the specific requirements of the biodiversity domain, but these requirements are not discussed.

To summarise, I think the paper is below the acceptance threshold but has the potential to be split into two contributions to be further refined: (i) a survey including a more detailed discussion of the different use cases, requirements and/or KG-construction methods implemented by different platforms, and (ii) a more detailed presentation of the iKNOW platform discussing the relevant requirements and design decisions for the biodiversity domain.

Minor remarks:
- the introduction is very confusing in my opinion trying to explain several terms from scratch (e.g, KG, reasoner, etc.) in a few sentences. Moreover, it is not clear to me why [1] is cited here.
- Some sentences may benefit a rewriting to make them less informal.
- "is needed that allows" (pag.8)

---

### Official Review · ~Julián_Arenas-Guerrero1 · 2022-04-04
**An abstract survey of KG platforms and a proposal that are not well connected**

**Rating:** 6
**Confidence:** 2

**Review:**

This paper first presents a survey on KG platforms and then proposes one for the biodiversity domain.

The identification of previous KG platforms available in the literature is valuable, however the platforms are presented vaguely. As specified in the document, the authors have relied on the descriptions presented in the respective papers (that can be quite abstract) and are presented in a way that it is difficult for the reader to see the differences among them (for instance additional tables similar to Table 1 could be added). More specific information would improve the paper, for instance, is RML used for the customized ETL processes in the KG platforms? is SHACL used for KG validation?. The common functionalities in KG management platforms divided in the four categories is a good contribution of the paper.

The presented platform, as the previous part, is presented vaguely. The schema refinement, triple augmentation or quality checking are mentioned without any further explanation. Also a comparison of the proposed KG platform with the ones previously surveyed is missing and why it is better suited for the biodiversity domain.

---

### Official Review · ~Semih_Salihoglu1 · 2022-04-05
**Review for What is needed in a Knowledge Graph Management Platform? A survey and a proposal**

**Rating:** 5
**Confidence:** 5

**Review:**

The paper proposes 2 main contributions. The first one is to survey 11 recent KG platforms by examining functionalities broadly in 4 different categories. The other one is proposing a KG platform, named iKNOW, tailored for biodiversity research.
Although the 2 contributions are related to each other, I would suggest the authors to separate the 2, because it is difficult to understand what the main purpose of the paper is. What is the research question the paper tries to answer to? Surveying existing KG platforms and proposing a domain specific KG seem completely 2 different contributions that may potentially be answers to different research questions.
I see 2 options to pursue for the authors to decide how to position their paper.
1)	Remove iKnow and only focus on the survey. In this case, the authors should provide more about the features of the existing KG platforms. Current version of the paper in terms of survey lacks technical contribution. The authors defined 4 categories in which many features are mentioned. However, it is not clear why only 4 of the chosen 11 KG platforms support data preprocessing which seems a necessary part of the KG generation pipeline. Instead of defining what these features are and which platforms have those features, I would strongly recommend to the authors to write a detailed section of how each platform handles each functionality. What are the similarities/differences between platforms for a particular functionality? Which ones are more preferable and why? Try to answer those questions in your survey and explicitly provide such research questions at the beginning of the survey to enlighten the reader about the purpose of the survey. On top of the above changes recommended, I would also suggest a user survey about these chosen platforms. Instead of 11, you can niche pick a sample of platforms and define multiple use-cases (e.g., one for each category) and conduct a user study and as a result, report user feedback on platforms for each category. With such an analysis, you can immensely improve the quality of the paper.
2)	Remove survey and only focus on KGs in biodiversity research. In that case, you need to explain/answer the following questions:
a.	what is the current state of the art in KG for biodiversity research? Motivate the problem better by providing more references and practical applications out of KGs in biodiversity research.
b.	why would someone need a KG platform tailored for biodiversity research? Why can’t one use existing KG platforms for biodiversity research? What are the advantages/disadvantages of existing platforms when it comes to using for biodiversity research? Are data different? Are there any other domain specific functionalities that need to be supported in such KG platforms? Does technical background of potential users of a KG platform for biodiversity research demand a different User Interface?
Due to reasons explained above, I believe the current version of the paper is not ready to be accepted. After addressing suggestions mentioned above, I also believe the paper may improve.

Review by: Dr. Arif Usta

---

### Decision · Program_Chairs · 2022-04-11

**Decision:**

Reject

**Comment:**

Dear authors,

Thank you for submitting your paper. Unfortunately we don’t accept your paper now in its current state. We refer to the reviews for suggestions on how you can improve your paper.

Kind regards
Organizers of the Knowledge Graph Construction workshop 2022